# MSCAT: A Machine Learning Assisted Catalog of Metabolomics Software Tools

**DOI:** 10.3390/metabo11100678

**Published:** 2021-10-02

**Authors:** Jonathan Dekermanjian, Wladimir Labeikovsky, Debashis Ghosh, Katerina Kechris

**Affiliations:** 1Department of Biostatistics and Informatics, Colorado School of Public Health, University of Colorado Anschutz Medical Campus, Aurora, CO 80045, USA; JONATHAN.DEKERMANJIAN@CUANSCHUTZ.EDU (J.D.); DEBASHIS.GHOSH@CUANSCHUTZ.EDU (D.G.); 2Department of Education and Research, Strauss Health Sciences Library, University of Colorado Anschutz Medical Campus, Aurora, CO 80045, USA; wladimir.labeikovsky@cuanschutz.edu

**Keywords:** metabolomics, database, text mining, open-source software, workflows

## Abstract

The bottleneck for taking full advantage of metabolomics data is often the availability, awareness, and usability of analysis tools. Software tools specifically designed for metabolomics data are being developed at an increasing rate, with hundreds of available tools already in the literature. Many of these tools are open-source and freely available but are very diverse with respect to language, data formats, and stages in the metabolomics pipeline. To help mitigate the challenges of meeting the increasing demand for guidance in choosing analytical tools and coordinating the adoption of best practices for reproducibility, we have designed and built the MSCAT (Metabolomics Software CATalog) database of metabolomics software tools that can be sustainably and continuously updated. This database provides a survey of the landscape of available tools and can assist researchers in their selection of data analysis workflows for metabolomics studies according to their specific needs. We used machine learning (ML) methodology for the purpose of semi-automating the identification of metabolomics software tool names within abstracts. MSCAT searches the literature to find new software tools by implementing a Named Entity Recognition (NER) model based on a neural network model at the sentence level composed of a character-level convolutional neural network (CNN) combined with a bidirectional long-short-term memory (LSTM) layer and a conditional random fields (CRF) layer. The list of potential new tools (and their associated publication) is then forwarded to the database maintainer for the curation of the database entry corresponding to the tool. The end-user interface allows for filtering of tools by multiple characteristics as well as plotting of the aggregate tool data to monitor the metabolomics software landscape.

## 1. Introduction

The need to list and categorize software tools used in different phases of data analysis is recognized in the metabolomics and bioinformatics communities [1,2,3,4,5]. Technological advances, both in instrumentation and computation, have allowed for more comprehensive and sensitive measurements of metabolites. This has driven the growth of metabolomics applications to study biological mechanisms, discover biomarkers, diagnose disease, and monitor treatment responses [6,7,8]. The diversity of applications is matched by a diversity of instrumental approaches, experimental designs, as well as the expansion of statistical and computational methods applied to the growing amount of generated and curated data from metabolomics studies. This translates to a large, complex, and expanding collection of software tools used in metabolomics [1,9,10,11,12,13,14,15,16,17,18,19,20,21,22]. Unfortunately, it also translates into the fragmentation of software and data resources, blunting the full advantage of the expanded capabilities of modern metabolomics research [2,5,23]. Further, as metabolomic data are to be integrated into multiple -omics datasets (e.g., multiomics or mixomics) or as a tool for systems biology, the problem of finding the right tools and data resources is compounded [24,25,26,27].

The size, complexity, and variety of the set of software tools applicable to metabolomics studies introduce some difficulty in delineating their optimal use and adoption as well as their incorporation into best practices for reproducible science and data reuse. Metabolomics researchers may not have dedicated bioinformatics support to help navigate the toolsets, let alone to iterate the development of analysis workflows [5]. Multiple efforts have sought to provide a reasonably up-to-date overview of the software tools being developed [1,14,16,17,18,19,21,28]. However, these have relied on either labor-intensive human curation of the most notable tools as they are released and/or on crowdsourcing of the most recommended tools available. The pace of metabolomics and software development combined with the already-mentioned diversity of the field itself means that these efforts cannot cover the entirety of the tool landscape, and they are bound to be quickly out of date. Inspired by the work being conducted in other areas of bioinformatics tool development [2,29], we propose that surveying and assessing the current landscape of metabolomics software tools with respect to their functionality, interoperability, and reusability can be a “force multiplier” for the metabolomics methods development community and for researchers in general. To that end, we have developed MSCAT, a curated database of metabolomics software tools that uses Machine Learning (ML) and text mining to automatically assist the curator by scanning the literature for publications describing metabolomics tools and identifying new metabolomics tools by name (Figure 1).

## 2. Results

### 2.1. Categorization of Software Tools

Given the large variety of instruments, approaches, and applications represented in metabolomics research, any survey or catalog of software benefits from a framework to categorize these tools. Different schemes have been explicitly and implicitly proposed (see Table 1) by previous reviews and surveys. Two basic aspects emerge as criteria for tool categorization: instrumentation used and step in the analysis pipeline. However, many differences emerge among the schemes in their details. Another pitfall of an a priori granular categorization of tools is that the structure may limit representation of the kinds of workflows possible in metabolomics studies. This is especially a concern as the discipline develops new approaches and refinements. Thus, when categorizing tools for MSCAT, we opted for a coarse categorization based on seven functionality categories: Pre-processing, Post-processing, Statistical Analysis, Annotation, Database, Visualization/Network, and Data/Workflow Management. Within the first four categories listed, each software tool may implement multiple specific methodologies or tasks (e.g., Pre-processing tools may be for peak detection, normalization, batch correction, etc.), and these are cataloged in a separate field for each of the first four functionality categories for the user to further filter the results in the main UI table or search for them in the Workflow Builder. Suite solutions (tools that function at multiple pipeline stages) are curated as belonging to multiple pipeline categories.

**Table 1 metabolites-11-00678-t001:** Categorization (rows) of metabolomics software tools in previous surveys (columns).

Kusonmano et al. [30]	Spicer et al. [1]	Stanstrup et al. [31]	Misra et al. [18]	Chang et al. [32]
**Data Acquisition** Instrument	**Pre-Processing** LC-MS GC-MS NMR	**Metabolite Profiling** MS Data ProcessingNMR Data ProcessingUV Data Processing	**Data Preprocessing**	**Data Preprocessing**
**Data Preprocessing** Normalization Identification Binning	**Annotation** Identification Quantification	**Metabolite Annotation** Ion grouping MS/MS Structure Databases	**Statistical Tools**	**Statistical Analysis**
**Data Analysis** PCA Clustering SVM PLS	**Post-Processing**	**Data Analysis** Statistics Network Analysis Pathway Analysis	**Annotation Tools**	**Identification**
**Data Interpretation** Databases Network Visualization Pathway Analysis	**Statistical Analysis**		**Biological Interpretation**	**Functional Analysis**
	**Workflow and other tools**		**Databases**	**Metabolic Modeling**
			**Instrument-based**	
			**Workflows**	

To supplement the stage-based categorization of tools, these are also curated by instrumentation category (LC-MS, GC-MS, MS, NMR) and by biomolecule type to accommodate tools that are used in multi-omics (i.e., not strictly the analysis of metabolite data). Further software tool characteristics (OS, programming language, version, user interface type, etc.) are curated to ensure “findability” of software and to be compatible with a simple software ontology (https://github.com/allysonlister/swo, accessed on 20 September 2021).

### 2.2. Database Design

We designed a relational database using PostgreSQL [33,34] (version 12.1) housing 17 tables that were normalized to the fifth normal form [35]. Following these normalization guidelines allows for a database that can be updated frequently while preventing data inconsistency and minimizing redundancy. The tables in the database are listed in Table 2.

We chose to base MSCAT on a relational database model rather than as a single flat table to allow for extensibility. The software tool’s name was chosen as the primary key or foreign key across tables, giving the data model a primarily one-to-many relation structure.

### 2.3. Literature Mining: Tool Publications

To begin populating our database, we defined a scope of the metabolomics tools to find and curate. As dictated by best practices, we limited our survey to open-source software tools. Additionally, in order to ensure both findability of documentation and to obtain rich information for tool curation, we limited ourselves to metabolomics software tools that have a publication associated with them (i.e., a publication announcing and describing a tool or its development). Thus we needed to mine the metabolomics literature to find a specific type of publication, also known as a Document Classification Problem. As a first approach, we built a PubMed query comprising the main concepts of metabolomics and software through keywords and Medical Subject Headings (MeSH) (Table 3).

The query defined by Table 3 returned 987 results as of 16 June 2021, and upon manual curation of titles and abstracts, 45.3% of them were publications of the desired type (i.e., publications primarily about a metabolomics software tool or tools, as opposed to metabolomics applications where a software tool is talked about in the abstract).

Manual checking of some previously logged publications with this literature search quickly revealed, however, that this simple query was not capturing all of the metabolomics software literature. We then formulated a second version of the PubMed query to cast a wider net. (Table 4).

This second query returns 9698 results (as of 16 June 2021), and upon manual curation of the titles and abstracts, 39.2% of them were publications of the desired type. This number is close to that obtained by the first query, however, it includes more software papers in areas outside of metabolomics, thus, while the second query catches more relevant papers, the “signal-to-noise” ratio is lower. As we will see in the following section, this may also affect the performance of our tool name identification method.

As mentioned above, the primary key in the MSCAT main database table is the tool’s name since multiple papers can talk about the same tool, and we want to be able to detect tools named anywhere in the metabolomics literature. Further, although FAIR (Findable, Accessible, Interoperable, and Reusable) Principles for Research Software [36,37] call for unique identifiers for software, adoption of this metadata standard is not complete. Thus, in order to properly track the tools to curate as they are published, in practice, we needed a way of detecting the tool name in the literature query results. In other words, on top of our literature query, we needed to solve a Named Entity Recognition (NER) problem [38,39].

### 2.4. Literature Mining: Tool Name Detection

Using the information that is retrieved by mining PubMed, our goal now was to detect metabolomics software tools contained within the text. For this reason, we employed a Named Entity Recognition model to extract predicted probabilities that a certain token, usually a word, is a metabolomics software tool name.

The title often contains information about the metabolomics tool name if the article is a true positive. Thus, we concatenated all titles with their respective abstracts obtained from the first PubMed query above. To train a NER model, we needed to first split the data into training and testing sets. We opted for an 80% for training and 20% for the testing split. To generate both the training and testing sets, we tokenized the words in the concatenated title and abstracts, then we removed English stop words, as defined by the Python Natural Language Toolkit (NLTK) package [40], and we tagged each token with a part of speech (POS) tag. We also identified where sentences end for each collection of tokens, and we created our label by tagging tokens that match a predefined list of known metabolomics software tools (combining [1] and our manual curation). An example of the processing can be seen in Table 5 (see Materials and Methods). Finally, we formatted the training and testing data to conform to the conference on natural language learning (CoNLL) format. We used a pre-trained deep learning model, provided by the Python spark-nlp package [41], to produce ELMo embeddings [42] for each sentence’s tokens. These word embeddings were used as features in the deep learning NER model. The deep learning (neural network) architecture that the NER model was built on starts with a character convolutional neural network (CNN) followed by a bi-directional long short-term memory (LSTM) layer followed by conditional random fields (CRF) layer [43,44]. The F1 score (see Materials and Methods) is an evaluation metric that aims to balance a model’s predictive correctness (precision) and its ability to identify the relevant data (recall). Against the dataset obtained from the first PubMed query, the CNN-LSTM-CRF model performs with an F1 of 76.5% versus an F1 of 64% using the CRF layer alone. When applied against a dataset from the second, broader PubMed query, the CNN-LSTM-CRF model performs with an F1 of 63.5% while using only the CRF layer performs with an F1 score of 45%. The poorer performance of the model on the second query may be partly due to the sparseness in tool name tokens in the dataset text. However, the size of the curated training set used to test the second query was also much smaller in relation to the size of the dataset (600 out of 9698 abstracts) compared to the curated training set used for the first query (600 out of 987 abstracts). Thus, this may also indirectly affect the NER performance. In the end, we chose to use the first, narrower PubMed query in the currently available version of MSCAT, and future updates will feature an optimized literature query.

### 2.5. Database User Interface

The current version of the MSCAT presents the user with three different views of the data (Figure 2). The user interface (UI) communicates with the PostgreSQL backend and is built using the Dash web framework. The main tab presents the tools and a subset of their characteristics sorted by how recently the tool has been updated and by the number of citations from the corresponding publication. The user is able to filter the table view by any combination of matching values in multiple fields. The user is also able to extract the full dataset of MSCAT as a single table in csv format from this tab.

In the middle tab of the application, we displayed aggregate data from the curated tools in MSCAT detailing the proportion of programming languages, methodologies, the timeline of release, and others. It is our intention to motivate from this view future detailed assessment of the software landscape (using the curated data in MSCAT) that may help identify gaps or overrepresentation of the kinds of metabolomics software being developed and areas where interoperability can be improved.

The third tab in the user interface is our first approach to providing the user with a way to design a metabolomics pipeline or workflow. In the current implementation, the user chooses specific methods within two or more functionality categories from among those most often linked together as pipelines or workflows (i.e., Pre-processing, Post-processing, Statistical Analysis, and Annotation) as well as other optional, additional discriminants (Instrument type, UI type, Molecule type) if needed. The interface then draws a Sankey-type diagram [45] representing the possible tool combinations that can constitute such a pipeline. This workflow generator interface does not present the other functionality categories (Database, Workflow Management, Visualization) since tools in those categories are largely agnostic or independent from the pipeline categories. The end-user can still perform a custom search of tools in those categories via the main interface tab.

### 2.6. Curation Workflow

We based our initial scan of the literature to start populating the database on the first query described above, plus previous reviews of metabolomics software tools [1,14,17,18,19,20,21,31]. From this scan of publications and NER identification of tools, we see that there are approximately 500 published metabolomics software tools (as of 16 June 2021), of which we have curated 400 thus far. For automation of updates of the landscape, MSCAT runs a script that calls the R functions that re-run the PubMed query and the Python code that detects the predicted tool names. The script then checks the updated results against previously predicted tokens, logged publications, and tools already in our database to prepare a report that it sends to the database curator with a list of possible new metabolomics tools (and their publication) to add. The script also runs an R function that queries the CrossRef API [46] for the citation numbers of each publication listed in MSCAT. Currently, we are running the literature mining script monthly and incorporating its results alongside the backlog of known tools still being curated. We are also using the ticket/issue feature of the code repository of MSCAT for community members to submit tool suggestions or data corrections.

## 3. Discussion

We have built MSCAT and its associated ML and automation parts in order to provide a catalog of metabolomics software tools that can be updated in a sustainable fashion without exclusive reliance on crowdsourced data or literature reviews. The database and its associated web interfaces are hosted at the Metabolomics Workbench website [47], an international repository for the metabolomics community. The catalog has three main purposes: make metabolomics tools more findable even if user requirements are very specific, providing a bird’s-eye-view of the efforts of the metabolomics software community, and, lastly, provide a way to determine interoperability between tools. Aside from the benefits to the user in finding and choosing compatible software for analysis workflows, our database project brings a structured framework to conduct surveys of metabolomics software tool development. We thus see the database also as a platform for the community to discuss needs and trends in the field and identify gaps in functionality as well as best practices for tool documentation and deployment.

The interoperability determination goal is represented in the current MSCAT implementation, as the workflow builder Table Currently, the determination of interoperability depends on salient features of the software tool (e.g., operating system dependencies, compatible file formats) and its classification of chief functionality (e.g., annotation, pre-processing, statistical analysis). Relying on this chiefly syntactic definition of interoperability is not error-proof, it depends strongly on user metabolomics expertise, and it does not adequately describe all the meaningful analysis workflows that can be assembled. We initially tried mitigating the lack of semantics by matching input and output file formats between tools in a pipeline. However, the file formats used in metabolomics do not provide a sufficient specification for this. The recent development of newer, more metadata-rich file formats [48,49,50] for metabolomics tells us there is an increasingly urgent need to standardize the knowledge representation and metadata elements in this community to achieve better interoperability and information exchange. There is a similar problem being worked on in the healthcare space [51], and a similar effort is likely needed when dealing with large, diverse datasets as in metabolomics and multi-omics. Future work will aim to build some semantic characterization of the software tools into their curation into MSCAT. This characterization could combine the metadata in the new file formats together with a Software Description Ontology [52] thus that future versions of the MSCAT can characterize software tools by their input and output variables to programmatically assess interoperability and outline all possible meaningful analysis pipelines [53]. We expect that this future capability would help improve the compliance of metabolomics studies to FAIR principles as well as provide an entry point for researchers with less specialized expertise to formulate a sophisticated data analyses workflow that takes full advantage of what the metabolomics software community has to offer.

## 4. Materials and Methods

We surveyed the landscape of available metabolomics tools by mining the literature repository PubMed. PubMed citations come from MEDLINE indexed journals as well as journals and manuscripts deposited in PubMed Central. We used the easyPubMed R package [54] to make API calls to PubMed, retrieving the abstracts that matched the PubMed query that we defined. The first PubMed query used for model training and initial scanning was:

((software [MeSH Terms] OR “programming language” [All Fields]) AND (metabolomics [MeSH Terms] OR “metabolomics” [All Fields] OR “metabolomics” [All Fields] OR “metabolomic” [All Fields] OR “metabonomic” [All Fields] OR “metabonomics” [All Fields])).

The second PubMed query, which collects many more publications but at a lower percentage of relevant ones, was:

(((“metabolomic” [All Fields] OR “metabolomics” [All Fields] OR “metabonomic” [All Fields] OR “metabonomics” [All Fields] OR “metabolonics” [All Fields] OR “metabolite” [All Fields] OR “metabolites” [All Fields] OR “multiomic” [All Fields] OR “multiomics” [All Fields] OR “mixomic” [All Fields] OR “mixomics” [All Fields] OR “metabolome”[All Fields] OR “metabolomes”[All Fields] OR Metabolomics[Mesh] OR metabolome [Mesh])) AND (((algorithm OR toolkit) AND (code OR software)) OR “open source” OR “source code” OR “web app” OR “web application” OR “command line” OR “programming language” OR (software AND (framework OR pipeline OR tool OR package OR suite OR workflow)) OR github OR gitlab OR sourceforge OR Bioconductor OR biopython OR biojava OR bioruby OR “Computing Methodologies” [Mesh])).

The mined data included the article’s PMID, the title, and the abstract. Publication types were curated manually using Endnote software (Clarivate).

We concatenated all titles with their respective abstracts. To train a NER model, we split the 600 manually curated abstracts retrieved from mining PubMed using Query 1 into training and testing sets. We opted for an 80% for training and 20% for testing split and further pulled 10% from the training as a validation set. To both the training and testing sets we tokenized the words in the concatenated title and abstracts, then we removed English stop words, as defined by the Python NLTK package, and we tagged each token with a part of speech (POS) tag. We also identified where sentences end for each collection of tokens, and we created our label by tagging tokens that matched a curated list of known metabolomics software tools derived from the 600 manually curated abstracts. An example of the processing is shown in Table 5. Finally, we formatted the training and testing data to conform to the conference on natural language learning (CoNLL) format. We then used a pre-trained deep learning model, provided by the Python “sparknlp” package [41], to produce ELMo embeddings for each sentence’s tokens. These word embeddings were used as features in the deep learning NER model that was generated using the “sparknlp” package.

The deep learning (neural network) architecture that the NER model is built on includes a character convolutional neural network (CNN) followed by a bi-directional long short-term memory (LSTM) layer followed by conditional random fields (CRF) layer [55]. The parameters of the character CNN take 100 number of characters, a filter size of 25, and a kernel of 3 × 3, and the LSTM cells are of size 128 for the bi-directional LSTM layer. We evaluate model performance using the f1 score statistic.
$$f1\;\mathrm{score}=2\frac{\left(\mathrm{precision}\right)\left(\mathrm{recall}\right)}{\mathrm{precision}+\mathrm{recall}}=\frac{\mathrm{TP}}{\mathrm{TP}+\frac{1}{2}\left(\mathrm{FP}+\mathrm{FN}\right)}$$
where TP, FP, and FN are the true positive, false positive, and false negative predictions, respectively.

## Figures and Tables

**Figure 1 metabolites-11-00678-f001:**
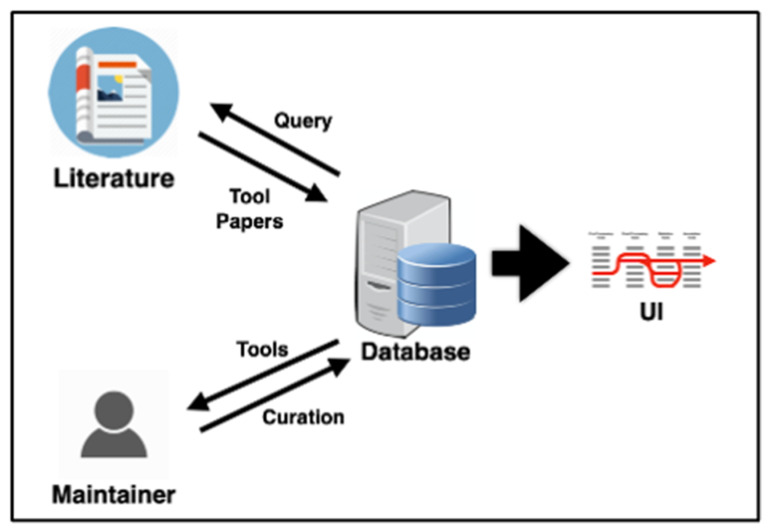
Schematic of MSCAT function. A semi-automated process is implemented where the database server periodically queries the literature and extracts new published tools to be displayed in the user interface (UI). The tools classification is then curated by the database maintainer.

**Figure 2 metabolites-11-00678-f002:**
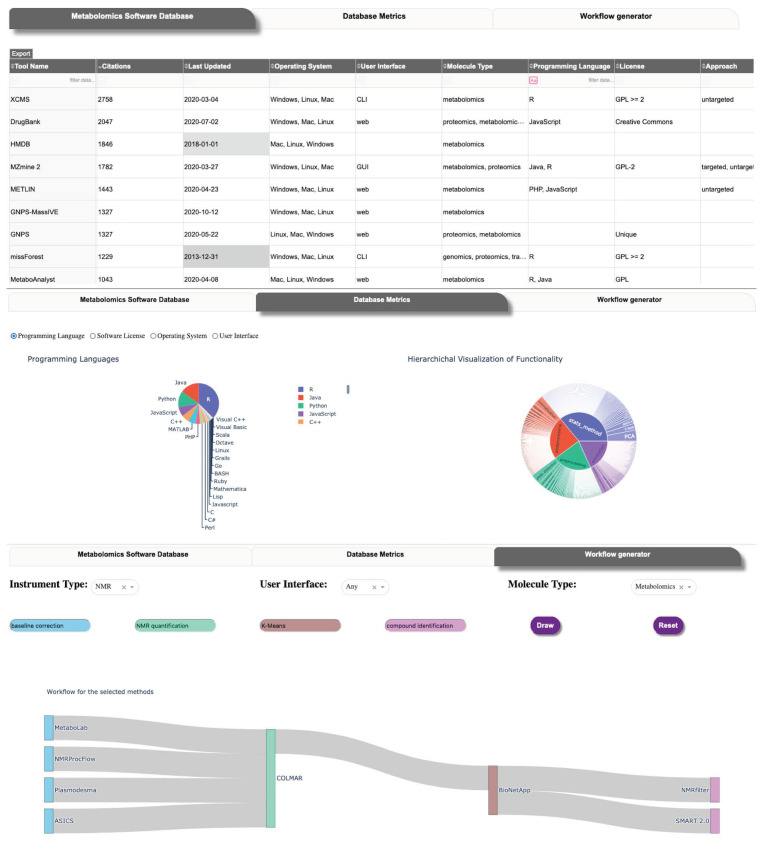
User interface of MSCAT. **Top**: main table view. **Middle**: aggregate tool data visualization. **Bottom**: workflow builder.

**Table 2 metabolites-11-00678-t002:** Structure of the MSCAT database consisting of 17 tables and their descriptions.

Database Table Name	Description
Main	Tool name, release year, link to website, date of last.update, etc
**Functional Characteristics**	
Molecule Type	Molecule types the tool works with
Approach	Metabolomics approach (targeted, untargeted) if defined
Instrument Data	Type of instrument data the tool uses, if defined
Functionality	Functionality categorie(s) of the tool
Annotation	Annotation methods the tool provides
Pre-processing	Pre-processing methods the tool provides
Post-processing	Post-processing methods the tool provides
Statistical Analysis	Statistical methods the tool provides
**Software Characteristics**	
File formats	The file formats the tool works with
Operating system	Tool’s operating system compatibility
Programming language	Tool’s available programming languages
Software license	Tool’s software licenses
User interface	Tool’s available User interfaces
Containers	Is the tool available as a container (URL)
Citations	Aggregate number of citations of tool publications
Publications	Publications associated with the tool

**Table 3 metabolites-11-00678-t003:** Outline of first PubMed query to obtain metabolomics software tools papers based on metabolomics and software-related keywords and MeSH terms.

Concept	Metabolomics	Software
**Keywords**	Metabolomics Metabolonics Metabonomics	Programming Language
**Controlled Vocabulary (Mesh)**	Metabolomics	Software

**Table 4 metabolites-11-00678-t004:** Outline of second PubMed query to obtain metabolomics software tools papers based on metabolomics and software-related keywords and MeSH terms.

Concept	Metabolomics	Software
**Keywords**	Metabolomics Metabonomics Metabolonics Metabolite Multiomic Mixomic Metabolome	Algorithm Toolkit Code Software (framework/pipeline/tool/package/suite/workflow) Open source Source code Web application Command line Programming language Github Gitlab Sourceforge Bioconductor Bio(python/java/ruby)
**Controlled Vocabulary (MeSH)**	Metabolomics Metabolome	Computing Methodologies

**Table 5 metabolites-11-00678-t005:** Example of processed training data. Where each row corresponds to a word from the collection of abstracts in the training data. The Part of Speech column (POS) tag tokens as a common noun (NN), adjective (JJ), etc. The sentence ID column identifies a training example (i.e., all tokens with the same Sentence ID are inserted into the model as one training example. The label column describes whether a token is a software tool (T) or not (O).

Token	POS	Sentence ID	Label
MetaComp	NN	1.0	T
comprehensive	JJ	1.0	O
analysis	NN	1.0	O
software	NN	1.0	O
comparative	JJ	1.0	O

## Data Availability

The code and data behind MSCAT are hosted in a version control repository (https://gitlab.com/metabolomics-tools-database/database-container, accessed on 20 September 2021).

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
