# Peer review of "MSCAT: A Machine Learning Assisted Catalog of Metabolomics Software Tools"

_metabolites, 2021, doi:10.3390/metabo11100678_

Round 1

Reviewer 1 Report

The authors introduce a computational workflow to select and organize software information in metabolomics from literature. The methods are understandable and I appreciate the hard manual work associated with this computation.

Major:

  • Literature mining is acceptable provided that the following manual curation is accurate. In the current dataset, many software programs lack information for file formats and the instrument data are not well organized. Please check the file formats and platforms for the registered data. Brukerk BI-LISA & BI-Quant-Ur is not a format, for example.
  • The same for Functionality (Preprocessing, Postprocessing, Statistical). I do not understand the relationship between these columns because Functionality includes data processing. Maybe these columns should be merged, given the current level of randomness.
  • The workflow generator does not work fine due to the missing / unorganized level of pre/post-processing methods. The GUI is wonderful but the authors should at least curate information for major metabolomics platforms.

Minor:

  • The search function in the web interface is better to be case-insensitive.
  • Table 1 is very hard to understand because rows are unaligned. Remove lines between rows and insert arrows vertically instead. Switching rows and columns are also ok but again, do not place different categories in the same column.
  • 5th NF is very strict; I do not think this strictness is necessary for maintenance but it's the matter of authors.
  •  

Author Response

Response to Reviewer 1 Comments:

Overall: "The authors introduce a computational workflow to select and organize software information in metabolomics from literature. The methods are understandable and I appreciate the hard manual work associated with this computation."

Response: we thank the reviewer for their thoughtful and useful comments. We have done some significant data harmonization of MSCAT to address their points and others as indicated below:

Point 1: "Literature mining is acceptable provided that the following manual curation is accurate. In the current dataset, many software programs lack information for file formats and the instrument data are not well organized. Please check the file formats and platforms for the registered data. Brukerk BI-LISA & BI-Quant-Ur is not a format, for example."

Response: we have corrected the incorrect file formats as well as resolved redundancy in the Instrument Data field. Further we have coalesced the different instrumentation options into fewer categories (NMR, LC-MS, GC-MS, MS). This is described in the new version of the manuscript (lines 77—93). We note here and in the acknowledgements that this recategorization and data quality check were done in consultation with metabolomic researchers. Part of the design of MSCAT contemplates and expects the continuous feedback from its users (submitted, for example, through the feedback link at the bottom of the page) for continuous improvement of the curation workflow and for corrections.

Point 2: "The same for Functionality (Preprocessing, Postprocessing, Statistical). I do not understand the relationship between these columns because Functionality includes data processing. Maybe these columns should be merged, given the current level of randomness."

Response: we have likewise recategorized the Functionality field to make the information clearer. Now the possible options for tool functionality now are: Preprocessing, Postprocessing, Statistical Analysis, Annotation, Database, Visualization (includes network analysis), and Data/Workflow Management. Tools can provide functionalities in multiple categories and are catalogued accordingly. In addition, the specific methods of the Preprocessing, Postprocessing, Statistics, and Annotation are listed in separate fields. See lines 77—90 in new manuscript version.

Point 3: "The workflow generator does not work fine due to the missing / unorganized level of pre/post-processing methods. The GUI is wonderful but the authors should at least curate information for major metabolomics platforms"

Response: the current workflow generator only covers tasks in 4 out of the 7 functionality categories (Preprocessing, Postprocessing, Statistical Analysis, and Annotation). Task in these 4 categories are most commonly linked together as “pipelines” and therefore the focus of the current workflow generator. Major platforms that provide multiple functionalities in these categories will show up as options in the drawn Sankey diagram. Users looking for platforms or tools with functionalities outside the above 4 categories (e.g. users specifically looking for Database or Visualization tools) can currently use the main UI tab and filter under the Functionality column. This is described in lines 203—213 of the new manuscript version.

Point 4: "The search function in the web interface is better to be case-insensitive."

Response: we have changed the code and interface to allow user to enable or disable case sensitivity for each field search.

Point 5: "Table 1 is very hard to understand because rows are unaligned. Remove lines between rows and insert arrows vertically instead. Switching rows and columns are also ok but again, do not place different categories in the same column."

Response: we have modified Table 1 according to this suggestion.

Point 6: "5th NF is very strict; I do not think this strictness is necessary for maintenance but it's the matter of authors."

Response: we reasoned that strict normalization will make maintenance and error detection easier, but foremost we chose strict normalization to also allow for extensibility in future iterations of MSCAT.

Reviewer 2 Report

I have the following questions for the authors -

  1. Is this a database or the users can run their new query?
  2. The database currently has tools from October 2020. What about the tools developed in 2021? How often do you update your database?
  3. How does it compare to using some popular search tools like Google? Can you please provide a matrix of time/results using comparable tools?
  4. How does your database compare to some other tools like https://github.com/RASpicer/MetabolomicsTools? This is an old publication but there are  several other review papers since this publication that list available tools.

Author Response

Response to Reviewer 2 Comments

We thank the reviewer for their thoughtful questions. We provide our answers below:

Question 1: "Is this a database or the users can run their new query?"

Response: this is a database where the end-user can perform queries by a combination of filtering and sorting in the fields displayed in the UI tabs. We may in the future provide an interface for end-users to perform SQL queries but we did not want to complicated the UX. We also provide a button on the main interface for data export (as a csv file) so end-users can obtain and query the data in MSCAT on their own.

Question 2: "The database currently has tools from October 2020. What about the tools developed in 2021? How often do you update your database?"

Response: our current workflow is set up to update the data released in MSCAT monthly. Currently, that involves curating new tools detected by our literature mining algorithm but also curation of older tools in our “backlog”. We expect that in the next few months, our historical tool backlog will be curated fully as part of the database and from then on the monthly updates will only cover newly released tools and any corrections of the data submitted by the users and maintainers. This is noted in lines 222—228 of the new manuscript version.

Question 3: "How does it compare to using some popular search tools like Google? Can you please provide a matrix of time/results using comparable tools?"

Response: a search in Google may identify some of these software tools but the results would not be organized in a user friendly manner or in a way that would allow comparison between different tools. MSCAT’s table interface allows the end user to select and narrow the tool attributes they are interested in and compare the attributes of subsets of tools side-by-side. We think the latter is necessarily more useful than using search engines to crawl links according to a set of keywords and then manually fishing the results into a list of tools that then need to be compared to each other, also manually.

Question 4: "How does your database compare to some other tools like https://github.com/RASpicer/MetabolomicsTools? This is an old publication but there are  several other review papers since this publication that list available tools."

Response: publications and resources like Spicer’s and Misra’s are extremely useful in the description and categorization of the better metabolomics tools and they have been an essential part for our initial building of MSCAT. One problem we saw with these previous efforts is that they became at best snapshots of the software landscape. Thus, part of the main motivations in our building of MSCAT was to provide a dataset that was regularly updated and that could be more flexibly queried as well as extensible. We acknowledge that this requires active maintainership and we have endeavoured to keep MSCAT modular, extensible, and portable (i.e. built around containers) in addition to have the code and data open so as to avoid a single point of failure from our side as maintainers.

Round 2

Reviewer 1 Report

The authors followed the comments sincerely and the manuscript became better. If possible, I suggest removing the keyword 'metabolonics' in Table 3 and 4. I could find only one article using this word in PubMed. 

This paper is only an introduction to a continuing effort of maintaining the website. I appreciate the hard work of the authors.

Author Response

Reviewer 1: "The authors followed the comments sincerely and the manuscript became better. If possible, I suggest removing the keyword 'metabolonics' in Table 3 and 4. I could find only one article using this word in PubMed. This paper is only an introduction to a continuing effort of maintaining the website. I appreciate the hard work of the authors.”

Response: We thank again the reviewer for their thoughtful comment and their correct appreciation of our project. We envision future refinement of the literature mining search strings as we accumulate publication data and as the field evolves. Having said that, to eliminate the metabolonics keyword from our search now would dictate that we re-run the search and check the classification of abstracts for the sake of reproducibility so we opt to keep it as is until the next iteration of our literature mining method.

Reviewer 2 Report

Thanks for your answers to my questions. If there is a huge backlog in your monthly updates (Ref: Q#2) then how does it not become a snapshot tool like others (ref: Q4)?

Author Response

Reviewer 2: "Thanks for your answers to my questions. If there is a huge backlog in your monthly updates (Ref: Q#2) then how does it not become a snapshot tool like others (ref: Q4)?”

Response: We thank their reviewer for their thoughtful comments on our project and process. The main reason the backlog is sizeable at the moment is that the project so far has concentrated on the structure, data definitions, data harmonization, and building of the database application. During this phase, population and curation of database records was necessarily slower. Having finished this initial phase, we as maintainer can now curate up to 10 tools per day. At this rate, the backlog should be cleared within ~3 months. Going forward, we have been tracking the rate of new tools being published and over the past few months the number of new and updated metabolomics software tools is about 20 per month which is well within the curation capacity mentioned above to keep the database updated.

Round 3

Reviewer 2 Report

Thank you for providing detailed answers to my questions. I would still like to see tangible results of backlog removed from the current speed.

Author Response

"I would still like to see tangible results of backlog removed from the current speed."

We are attaching a plot of database modification over time. The plot shows that rate of tool curation increases over time as the database design and building is finalized. We are now at 400 tools in the database and the approx. 100 tools remaining in the backlog will be done over the next week.

We also note that our literature search as well as the work of Misra show that the number of new or majorly updated metabolomics tools released each year is fewer than 100. Thus, the changes of the metabolomics software landscape are well within our capability or curating approx. 10 tools per day.
